# Epidemiological Trends of Haematological Malignancies in Belgium 2004–2018: Older Patients Show the Greatest Improvement in Survival

**DOI:** 10.3390/cancers15174388

**Published:** 2023-09-01

**Authors:** Kris Henau, Tim Tambuyzer, Bart Van Gool, Liesbet Van Eycken, Hélène A. Poirel

**Affiliations:** Belgian Cancer Registry, Rue Royale 215, 1210 Brussels, Belgium; kris.henau@kankerregister.org (K.H.); tim.tambuyzer@kankerregister.org (T.T.); bart.vangool@kankerregister.org (B.V.G.); elizabeth.vaneycken@kankerregister.org (L.V.E.)

**Keywords:** haematological malignancies, epidemiological trends, population-based cancer registry, older patients

## Abstract

**Simple Summary:**

This unique nationwide population-based study analyses the epidemiological trends of 24 main types of haematological malignancies (HMs), including in children, by age (Belgium, 2004–2018). Particularly in the older population, increased incidence and survival are likely explained by the lagged spread of diagnostic and therapeutic innovations over the last two decades. This “real-world” study is useful to better monitor the impact of therapeutic changes in HMs at the population level.

**Abstract:**

(1) Background: Haematological malignancies (HMs) represent a heterogeneous group of mostly rare cancers that differ in pathophysiology, incidence, and outcome. (2) Methods: Our study aims to understand the epidemiological situation and trends of 24 main types of HMs in Belgium over a 15-year period, with a focus on the impact of age. Age-standardised incidence, average annual percentage change (AAPC), 5- and 10-year relative survival (RS) and RS trends were estimated for all HMs (N = 94,415) diagnosed between 2004 and 2018. (3) Results: Incidence rates of HM increased, mainly in the 70+ age group (AAPC: 3%). RS varied by age and HM type. For each HM type, outcome decreased with age. The greatest decrease with age in 5-year RS is observed for aggressive HM, acute myeloid leukaemia (AML), acute lymphoblastic leukaemia, and Burkitt lymphoma, from 67%, 90%, and 97% below 20 years, to 2%, 12%, and 16% above 80 years of age, respectively. The moderate improvement in 5-year RS over the 2004–2018 period for all HMs, of +5 percentage point (pp), masks highly heterogenous outcomes by HM type and age group. The most impressive improvements are observed in the 80+ group: +45, +33, +28, and +16 pp for Hodgkin lymphoma, immunoproliferative disorders, follicular lymphoma, and chronic myeloid leukaemia, respectively. (4) Conclusions: The increasing incidence and survival over the 2004–2018 period are likely explained by diagnostic and therapeutic innovations, which have spread to populations not targeted by clinical trials, especially older adults. This real-world population-based study highlights entities that need significant improvement, such as AML.

## 1. Introduction

Haematological malignancies (HMs) are relatively common and account for 11% of the total cancer incidence in Belgium. The incidence and mortality rates of HMs are close to those of colorectal cancer [1]. However, HMs are a heterogeneous group of cancers, originating from different cells of the bone marrow or the lymphatic system. They differ in incidence, physiopathology, prognosis, and survival. More than 150 malignancies are defined in the 2017 WHO classification [2], and most are considered rare cancers according to the definition of RARECARENet [3].

In the past few decades, there has been major progress in understanding the biology of HMs, driven by novel technologies such as gene expression profiling and next-generation sequencing [2]. This has led to more objective and reproducible diagnoses of entities, relying on the physiopathology and/or the cell of origin, using multidisciplinary characteristics (morphology, immunophenotyping, genetics and clinical characteristics), thereby providing a more sound basis for the clinical care of patients. This has resulted in the design of a new generation of anti-cancer treatments that target the abnormal cells. These pivotal innovations, initially developed for HMs (the first use of monoclonal antibodies to treat mature B-cell lymphoid neoplasms was in 1997 and the first treatment of chronic myeloid leukaemia with a kinase inhibitor was in 1998), gave rise to growth in “precision medicine”.

“Precision diagnostics” has, however, also greatly increased the complexity of classification and registration procedures, which have as a result changed significantly over time. In 2001, the WHO published the first true worldwide consensus classification of HMs [4]. It is considered a founding breakthrough in the international harmonisation of characterisation and terminology of the different HM types. A revised version of this classification was published in 2008 [5]. In 2009, a group of experts from different European cancer registries created a coding manual for HMs in an effort to facilitate the use of the classification system (Haemacare manual [6]). Since then, there have been significant improvements in the definition of current as well as newly introduced entities. The growing importance of genetics in the identification of HMs is highlighted by the 43 malignant types (26%) defined by gene alterations in the 2017 WHO classification [2].

Population-based cancer registries provide a real-world picture of overall outcome that is complementary to the data from randomised clinical trials (RCTs), which rely on selected populations, usually excluding patients with advanced age and/or with comorbidities and/or with prior history of other malignancies. However, measuring the burden of the different types of HM and assessing the impact of precision medicine on outcome is a real tightrope walk for population-based cancer registries, as data must be consistently classified over the entire study period, while yet remaining up-to-date to make clinical sense.

Whereas the epidemiology of large, heterogenous and/or incomplete groups of HMs has been characterised using population-based cancer registry data (such as ECIS—European Cancer Information System—https://ecis.jrc.ec.europa.eu, accessed on 1 July 2023; SEER—Surveillance, Epidemiology, and End Results Program—https://seer.cancer.gov/, accessed on 1 July 2023), very few detailed and recent datasets that would allow for comparisons between all of the different HMs by age group are available at the national level [7,8,9,10,11].

The goal of our study is to provide a detailed description of the epidemiological situation and trends of the 24 main types of HM in Belgium over a recent 15-year period, with a specific focus on the impact of age at diagnosis, in order to guide clinical decision making and identify gaps for future research.

## 2. Materials and Methods

The Belgian Cancer Registry (BCR) is a national population-based cancer registry, collecting data since the incidence year of 2004 via two data sources: oncology care programs and pathology laboratories. The completeness and validity of the BCR data is regularly evaluated [1,12,13]. Highly trained employees revise all incoming data using an extensive set of automated and manual validation procedures based on the IARC guidelines [14] to ensure high quality of the data. If needed, the data source is consulted to provide additional details for cases with uncertain diagnosis, insufficient or erroneous data, or conflicting information.

For this study, all HMs (N = 94,415) diagnosed between 2004 and 2018 were extracted from the BCR database. Follow-up of patients was available up to the 1st of July 2020. HM was defined with the following morphology codes of the ICD-O-3.2: (9590–9993)/3 + (9751–9753, 9740–9741)/1. This group was further subdivided into HM types based on the Haemacare and WHO classifications, as well as the ICD-O-3 codes consistently available over the period 2004–2018 [1]. In order to accommodate progressive insights into the classification of mast cell neoplasms and Langerhans cell histiocytosis, and the corresponding changes in behaviour in the consecutive editions of the WHO classification, a selection of subtypes that were formerly considered malignant but are today considered borderline malignant were included. As registration of borderline malignant HMs is recommended but not mandatory at the BCR, trends in the number of these registrations were checked to detect a potential impact over time. We used the one-digit code for differentiation (B-cell vs. T- or NK-cell origin) in addition to the topography (bone marrow and blood for leukemic lymphoid neoplasms versus other for lymphomas) to better sub-classify cases registered with the poorly specified code 9591/3.

Here, we focus on the 24 main HM types, which represent 98% of all HM cases diagnosed between 2004 and 2018. A total of 2213 unclassifiable or unspecified lymphoid neoplasms (2090 other lymphoid neoplasms and 123 other types of leukaemia) were not included in this analysis (Table 1). The selection of International Classification of Diseases for Oncology 3rd edition (ICD-O-3) codes is detailed in Appendix A.

For each of the 24 entities as well as the whole group of HMs, we calculated crude and age-standardised incidence rates using the world standard (WSR) population (N per 100,000 person years) as well as the European standard (ESR) population (1976 and 2013) to allow for international comparisons. Trends in age-standardised incidence (WSR) over the study period were evaluated by calculating the average annual percentage change (AAPC) with the corresponding 95% confidence intervals (95%CI) [15].

Five- and ten-year RS with corresponding 95%CI were estimated using the Ederer II methodology [16]. In addition, we included conditional 5-year relative survival up to 5 years after diagnosis. This is the relative survival proportion given that the person has already survived the first X years since diagnosis (results are shown for x = 1 to 5 years).

For this publication, we chose to mainly stratify results by age group rather than sex, in order to have sufficient cases per age category and HM type for the analyses of incidence and survival, in particular trends over time. However, results stratified by sex are available [1].

## 3. Results

### 3.1. Incidence

HMs represent a wide variety of malignancies. Following the RARECARENet [3] definition of rare cancers (crude rate (CR) of 6/100,000 person-years), only plasma cell neoplasms (PCNs), diffuse large B-cell lymphoma (DLBCL), MDS, and CLL/SLL can be considered “common”. With a CR of 5.4, AMLs are close to the cut-off of 6. However, AMLs are subdivided into at least 22 different subtypes, all of which are considered “rare” [2], as are all other HM types with CRs below 4 (Figure 1).

HMs are mostly diagnosed in older adults with an overall median age of 70 years (Figure 2, Table 1). However, HM is the most frequently diagnosed malignancy in younger patients. Although HMs diagnosed before 20 years of age account for less than 3% of all HMs, they represent nearly half of all new cancers diagnosed in this age group (Appendix A).

Although almost all HM types can be diagnosed at any age, the majority are predominantly diagnosed in adults with an incidence that increases with age. Only ALL/LL, HL, histiocytic, and dendritic cell neoplasms (HDCNs), Burkitt lymphoma/leukaemia (BL) and mast cell neoplasms (MCNs) exhibit a bimodal distribution, with two incidence peaks in younger and older patients [1] and a resulting median age below 50 years (Figure 2, Table 1).

ALL/LL is the most frequently diagnosed HM (46%) in children (0–14 years), followed by HL, which is the predominant HM in adolescents (15–19 years) and young adults (20 to 39 years) (46% and 33% of all HM diagnosis, respectively). From the age of 40, the most common HM types are PCN (15%), DLBCL (12%), CLL/SLL (12%), and MDS (12%) without any predominance of one type (Figure 3).

Most HMs show a higher incidence in males compared to females, with a higher M/F ratio for hairy cell leukaemia (HCL) (8.1) and mantle cell lymphoma (MCL) (4.2), except for essential thrombocythemia (ET) (0.7) and MCN (0.6) (Table 1).

### 3.2. Incidence Trends

Over the 2004–2018 period, the age-standardised incidence rates (WSR) of HM have increased in Belgium (AAPC: 1.2% with 95%CI [1.0%; 1.5%]). This is mainly observed in the older population (aged 70+), while the rates for children have remained stable. Large differences are observed between the different HM types. The largest increase (>5% annually) is observed for *BCR::ABL1*-negative chronic myeloid neoplasms (PV, ET, PMF) and for MCN. The WSR for MDS, myelodysplastic/myeloproliferative neoplasms (MDS/MPNs), PNK/TCL, and marginal zone lymphomas (MZLs) reveal an annual increase between 2% and 4%. The increased incidence of MDS/MPN predominates for chronic myelomonocytic leukaemia with an AAPC of 6% in males and 9.1% in females. In PNK/TCL, the increased incidence over time appears mostly in the leukemic presentations, with an AAPC of 10.8% in males and 9.3% in females. This group is mainly represented by T-cell large granular lymphocytic leukaemia (80%). The incidence trend for the other HM types remained stable or increased slightly over the period 2014–2018 (Table 2).

### 3.3. Survival

The 5- and 10-year RS for all HMs combined are 68.7% and 59.8%, respectively. Large differences in survival are, however, observed between the different HM types. Five-year RS is above 90% for HCL, PV, ET, MCN, FL, and CLL/SLL. The lowest RS is observed for AML with a 10-year relative survival that drops to 22% (Figure 4). The majority of patients (90%) who survived the first year after being diagnosed with AML are alive 5 years later (conditional 5-year RS; Appendix A).

We can also distinguish between certain HM types that have a nearly stable RS between 5 and 10 years of follow-up, such as HCL and BL (and to a lesser extent, ALL/LL and AML), versus others with a drop of over 33% between 5- and 10-year RS (PMF, MDS, MDS/MPN, PCN) (Figure 4). This is also illustrated with the conditional 5-year RS, which is around 100% 5 years after the diagnosis of HCL or BL, followed by HDCN, HL, pCTCL, MCN, and ALL/LL, while it is below 68% for PMF, MDS, MDS/MPN, and PCN (conditional 5-year RS; Appendix A).

For every HM type, survival outcome progressively worsens with age at diagnosis. The greatest variation is observed for BL, ALL/LL, and AML: the 5-year RS drops from 97%, 90%, and 67%, respectively, for children and adolescents, to 16%, 12%, and 2%, respectively, in 80+ adults. ET and PV exhibit a minor variation: the 5-year RS remains around 90% for the oldest age group of 80+ (Figure 5).

### 3.4. Survival Trends

The 5-year RS of all types of HM increased in recent years by 5 percentage points (pp), from 65% in 2004–2008 to 70% in 2014–2018 (Figure 6). The greatest improvement is seen in IPD (+11 pp), followed by CML (+8 pp), ALL/LL, CLL/SLL, and PCN (+8 pp). While all the mature lymphoid neoplasms show an improvement over time, no clear increase is observed for the myeloid neoplasms: AML, MDS, MDS/MPN, and *BCR::ABL1*-negative MPN (PV, ET, PMF), with the exception of CML.

The analysis based on age reveals larger differences in the survival improvement over time (Table 3). Across all HMs combined, patients over 60 years of age show the greatest increase in survival (+8 pp), while in children this is only +3 pp. However, the overall outcome in children is still far superior to that in adults (the 5-year RS during the period 2014–2018 reached 92% for the age group 0–14y versus 47% for 80+).

It is noteworthy that in children, 4 out of the 5 main HMs reached a 5-year RS above 90% during the period 2014–2018 (100% for BL, HL, and HDCN and 93% for ALL/LL) and that an important increase (+17 pp) is observed for AML, where there is still room for improvement. The increase for AML is less pronounced in young adults (+7 pp) and very limited in the older population. In contrast, improvement is much less pronounced in children for ALL/LL, while adolescents show a large improvement (+20 pp), with survival catching up with that of children, as well as young adults (+17 pp). Although notable progress is also detected for ALL/LL in the older age groups (+22 pp in patients between 60 and 79), the 5-year RS remains very low (37% over the period 2014–2018).

Young adults 20–39 show the largest improvement for BL (+21 pp), with survival progressing towards the level observed for children. PNK/TCL and MDS show a larger increase (+14 pp and +13 pp, respectively) in adults 20–59.

The most impressive increases are observed in the oldest age group (above 80), with more than +5 pp in 12 different HM types. The most pronounced increments of 5-year RS are observed for HL (+45 pp from 21% in 2004–2008 to 66% in 2014–2018), followed by IPD (+33 pp), FL (+28 pp), CML (+16 pp), PV (+15 pp), PMF (+13 pp), CLL/SLL (+12 pp), and MCL (+11 pp) (Table 3).

## 4. Discussion

This unique real-world population-based study of the 24 main types of HM by age, including children, evidences the spreading benefit of improved management of most HM types over a recent 15-year period at a national level, with a major impact in the oldest population.

International comparison of epidemiological data between population-based registries is a real challenge due to the risk of several biases: variations in selection criteria, different reference populations and modelling used for standardisation of incidence or survival rates, changes in classification over time, and discrepancies in the applications of coding recommendations. Some international organisations, such as the European Cancer Information System (ECIS) (https://ecis.jrc.ec.europa.eu, accessed on 1 July 2023) and the Global Cancer observatory (GCO) of the International Agency for Research on Cancer (IARC) (https://gco.iarc.fr/today/, accessed on 1 July 2023) publish data on large groups of cancers, with HMs divided into only four main categories that contain heterogeneous entities (HL, NHL, PCN, leukaemia). The U.S. Surveillance, Epidemiology, and End Results (SEER) registries provide statistics adjusted to the 2000 U.S. standard population for 10 groups of HMs [17]. Depending on the coding recommendations of four institutions, Adzersen et al. showed a variation of more than 10% in the extraction of incident lymphoid neoplasms [18].

### 4.1. Incidence

Most of the WSRs observed in 2018 in Belgium remain within the same range as those estimated in “Southern Europe” (ECIS definition—https://ecis.jrc.ec.europa.eu, accessed on 1 July 2023) in the same year. The most comparable data at the national level come from the report of the French network of registries (FRANCIM) over the period 1990–2018 [7]. The WSR for 2018 are projections from 2015 based on registries covering 20% of the French population. The major discrepancy is the higher incidence of MZL in France as compared to Belgium. MZL is a model of an antigen-driven malignancy, where epidemiologic risk factors partly drive lymphomagenesis. Differences in the geographic incidence of MZL have been attributed to modifications of predisposing infectious conditions, such as Helicobacter pylori and hepatitis C virus, the latter being more prevalent in Southern countries [8].

### 4.2. Incidence Trends

It is easier to semi-quantitatively compare the direction of trends between population-based registries, rather than the standardised incidences (which are often differently estimated). First, improved registration over time is highlighted in this study, as well as in the Spanish REDECAN registry [1], by the decrease in non-specific HMs. In Belgium, the main increases over the 15-year period are observed in the chronic myeloid malignancies, particularly the three main subtypes of *BCR::ABL1*-negative MPN (PV, ET, and PMF, which exhibit a similar AAPC of around 6%), followed by MDS and MDS/MPN. Since 2005, several somatic mutations have been identified in MPNs, e.g., in JAK2, MPL, and CALR [19]. These molecular biomarkers have led to updated diagnostic criteria that are more precise, greatly improving the differential diagnosis between a clonal myeloproliferative neoplasm and a reactive condition in the case of thrombocytosis or polycythaemia. This trend is also observed in Spain over the period 2002–2013 [9] and in Sweden [20], but not in French [7] or US [21] networks of registries over the periods 2003–2015 and 2002–2016, respectively. Variations in the coverage of the different cancer registries and the use of different standard populations make comparisons between studies difficult, however.

The Belgian population is ageing. This demographic characteristic will have a profound impact on the incidence of HM [22]. Moreover, a markedly increased incidence of MDS, MDS/MPN, AML, and PCN is observed in the older population (70+) in Belgium [1]. This may be explained by a more systematic bone marrow cell analysis, with molecular studies in the case of an abnormal blood cell count, especially in the older population, in order to obtain not only a diagnosis, but also a prognosis and guidance for treatments. We cannot exclude that MDS may have been under-registered in the earlier years due to the late application of classification changes (from non-malignant in ICD-O-2 to malignant neoplasm in ICD-O-3). The trend towards an increase is also observed in Spain [9], but not in the French [7] or US [21] registries, where incidence trends remain stable (and even decreased for MDS in the US after a peak in 2010). Pulte et al. have suggested that diagnosis of MDS is being made at a more advanced stage in the course of the disease in the US [23].

The major increase in incidence detected in Belgium for lymphoid neoplasms (T-cell large granular lymphocytic leukaemia and MZL) is also identified in the network of French registries [7]. Discordant trends are, however, observed for FL (incidence tends to increase in France while it remains stable in Belgium) and for CLL/SLL and IPD (incidence tends to decrease in France while it remains stable for CLL/SLL and increases for IPD in Belgium). For HL, a small increase is observed in Belgium and in France [7], while a decrease is observed in the US [24]. These minor variations may, however, be attributable to the imprecision of estimates and/or differences in the observation period.

The major increase in incidence of MCN should be relativised to the important changes in the classification over time. The increase is mostly due to better diagnosis and registration of systemic mastocytosis (neither indolent or malignant systemic mastocytosis were distinguished before the first revision of ICD-O-3 in 2012).

### 4.3. Survival

Survival data are often presented for only one adult age group, without respect to the potentially high variation in prognosis within this large age range [22,25]. Our study highlights the fact that the outcome of HMs varies considerably, not only by type of HM (Figure 4) but also by age at diagnosis (Figure 5). For each type of HM, we chose to divide the older population into two groups that were as homogeneous with the previous analyses as possible, i.e., 60–79 and 80+, even if clinicians may suggest age limits specific to HM type. We also included the younger population of children and adolescents to account for entities that have bimodal incidence with very different outcomes. Depending on the type of HM, adolescents (and even young adults) may benefit from being managed as adults or as children.

Very few such studies with detailed HM types are available for the period included here. Again, the most comparable population-based study has been reported by the French network of registries (restricted to patients aged 15+). The comparison of rate variations of different estimators should of course be interpreted with caution. The decrease in the 5-year RS with age by HM type observed over the 2009–2018 period in Belgium (Figure 5) remains within the same range as the variation observed for the same 22 HM types over the 2010–2015 period in France, using 5-year net survival with age standardisation based on multidimensional penalised splines [7,26,27,28]. In both countries, the larger variation with ageing among adults is observed for the more aggressive HMs, i.e., AML, ALL/LL, and BL, with a smaller gap for indolent HMs, PV, and ET, whose 5-year RS rate in the 80+ age group remains above 83%. Survival rates also remain broadly similar to those observed in the Nordic countries over the 2016–2020 period [29].

### 4.4. Survival Trends

The moderate improvement (+5 pp) in the 5-year RS over the 2004–2018 period for all HMs combined disguises significant heterogeneity in outcome by HM type and by age group (Figure 6). Outstanding improvements are observed in the oldest population, 80+, for several mature lymphoid neoplasms and, to a much lesser degree, for chronic myeloid neoplasms. In particular, due to the large increase in the 5-year RS observed for HL (+45 pp), IPD (+33 pp), and FL (+28 pp), the prognosis for the 80+ group approached that for adults aged 60–79. Major improvements are also detected for adults aged 60–79 for ALL/LL (+22 pp) and HCL (+18 pp). The 5-year RS of HCL reached the level of younger adults in the period 2014–2018.

These improvements in survival likely relate to advances in the management of treatments according to the frailty of older patients, improved management of toxicities, and the application of therapeutic schemes optimised within RCT in younger adults, to older populations. Additionally, access to innovative drugs in earlier therapeutic lines, especially the monoclonal antibody anti-CD20 (Rituximab) in most B-cell mature lymphomas and tyrosine kinase inhibitors in CML, were progressively introduced in the early 2000s [17,19,21,22,24,25,30,31,32,33,34]. The application of these innovative therapies likely lagged in older patients compared to younger adults. It is interesting to note that the study of the Spanish network of registries (REDECAN), which focused on an earlier period (from 2002–2007 to 2008–2013), found that survival improvements for most cancer groups (except acute lymphoid leukaemia) were greater for patients younger than 75 years than for older patients [35]. There is important heterogeneity amongst older patients, in comorbidity, coexisting conditions, socioeconomic status, and general level of functioning. However, the population-based studies show that they can benefit from curative-intent regimens and innovative drugs if correctly selected [22,25,33]. The tools currently used to assess a patient’s suitability for treatment, such as performance status, are insufficient, and a more adequate method of stratification needs to be developed [36].

Outstanding improvements are also detected in the younger population, mostly for the most aggressive HM (acute leukaemia and BL): AML in children (+17 pp), ALL/LL in adolescents and young adults (+20 pp and +17 pp, respectively), BL in young adults (+21 pp), and, to a lesser extent, PNK/TCL and MDS in adults only. This population can withstand more intensive treatments, is eligible for transplantation in relevant indications, and is more often a candidate for innovative drugs and regimens in the framework of randomised clinical trials.

For DLBCL, relative survival slightly increased between 2004–2008 and 2009–2013, and no further improvement was seen for the more recent period of 2014–2018 [1]. The major improvement was because of the introduction of the new monoclonal antibody (Rituximab) at the beginning of the study period, in the early 2000s [31]. This milestone was followed by smaller improvements in the following decade in all age groups [7,24,27,31,33]. However, promising therapies (new monoclonal and cell therapies such as chimeric antigen receptor T cells) [33] offer hope for improved survival in the coming years.

Despite a survival increase in the youngest patients, one of the most obvious areas for clinical improvement is in adult AML patients, who showed a small to no significant improvement in outcome; the 5-year RS in this age group remains the lowest of all HMs in our study, as well as in other population-based registries over a similar period [7,22,25,28].

Subject to the use of different survival rates, the trends observed in the FRANCIM study evolved in the same direction for all 22 HM types over a fairly similar period (2005–2015) [7,27,28]. The main discrepancy with the US network of cancer registries (SEER) is the overall decrease in the 5-year RS of MDS, but this was associated with an increased RS in younger (20–59) patients who are transplant eligible. Pulte et al. [23] suggest that changes in the classification and later diagnosis of MDS, coupled with no real advances in treatment options over the study period, could explain the global decrease in the survival rate.

### 4.5. Strengths and Limitation of the Study

A nationwide population-based cancer registry is a useful instrument to investigate cancer burden as well as to demonstrate how pivotal findings of clinical trials are implemented in routine clinical practice and affect outcomes in the general patient population over time. These studies are performed on an unselected collection of patients, including those less prone to participate in clinical trials (unfit, older patients, as well as those with comorbidities or prior history of malignancies). The classification of HMs is complex and most global reports on HMs are based on large groups comprising very heterogeneous HM types. Our study is one of the rare recent studies that analyses this heterogeneity in detail by subdividing it into 24 HM types and six age groups, including children.

Despite these strengths, some limitations of our study should be kept in mind. The changes in the international classification of HMs over the 2004–2018 period make it more challenging to accurately report specific HM types. The Belgian Cancer Registry lacks information on potential prognostic indicators and exposures that could allow deeper trend analyses by HM type. Finally, the relatively small number of patients in some HMs means higher uncertainty of incidence and survival estimates in certain subgroups.

## 5. Conclusions

The increase in both incidence and survival of HMs over time is likely explained by diagnostic and therapeutic innovations over the past two decades, which have gradually spread more widely to populations not targeted by clinical trials, especially the oldest population. This real-world population-based study also highlights entities that need significant improvement, such as AML.

These “real-world” population-based data are useful not only to better inform about HM burden but also to help (i) haematologists to better address profiles that urgently need improvement, such as AML, and (ii) policy makers to plan future health-care services and monitor the impact of therapeutic changes at the population level.

## Figures and Tables

**Figure 1 cancers-15-04388-f001:**
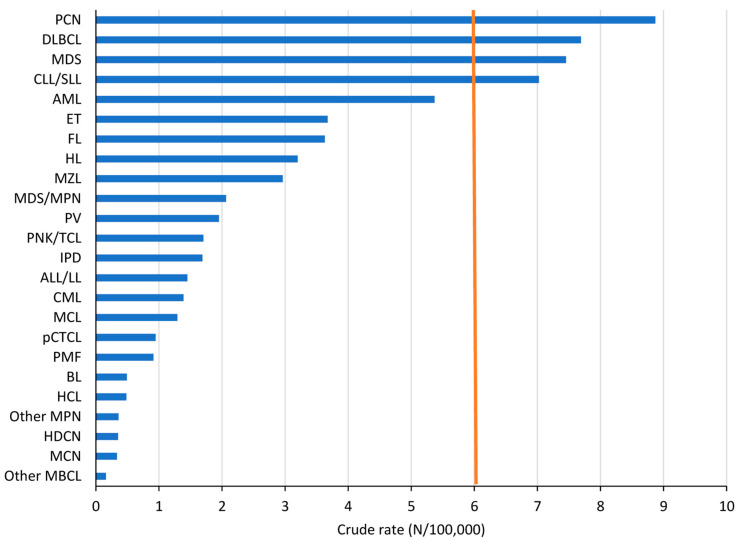
Incidence (crude rate) by HM type, both sexes combined, Belgium 2018. The brown vertical line shows the crude rate limit below which cancer is defined as rare according to RARECARENet3.

**Figure 2 cancers-15-04388-f002:**
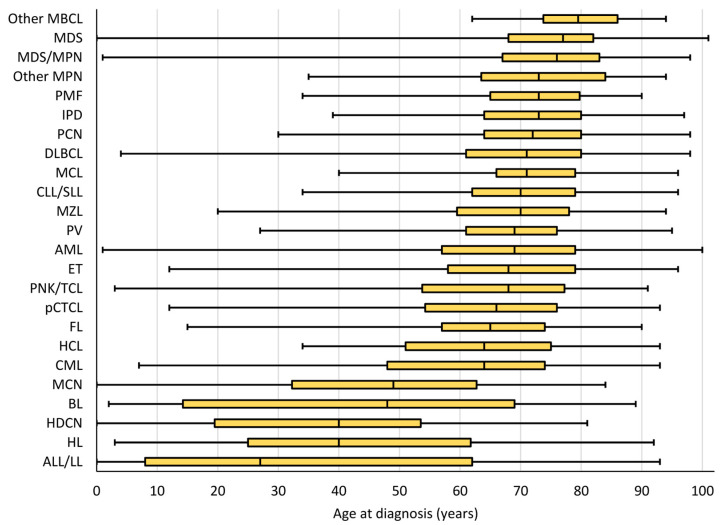
Boxplot ordered by median age at diagnosis, with interquartile range, minimum, and maximum age at diagnosis by HM type, both sexes, Belgium 2018. Each yellow box represents the interquartile range with the first quartile (Q1, left side), the median (Q2, vertical line within the box), and the third quartile (Q3, right side). The minimum and the maximum are represented at the left and right extremities, respectively, of the line which extends the box.

**Figure 3 cancers-15-04388-f003:**
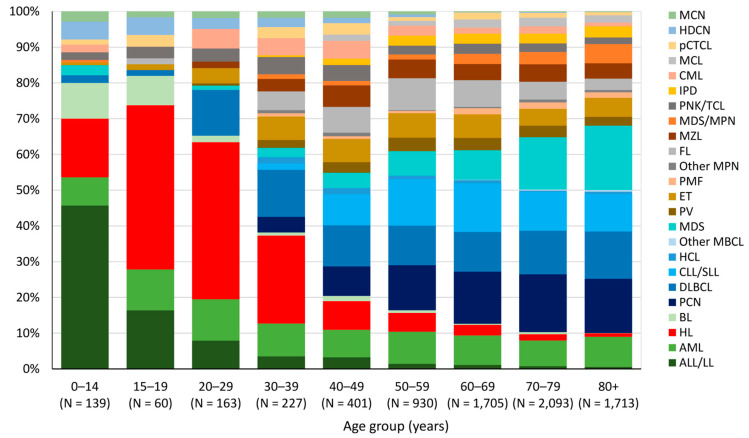
Incidence by HM type and age group, Belgium 2018. N: Absolute number.

**Figure 4 cancers-15-04388-f004:**
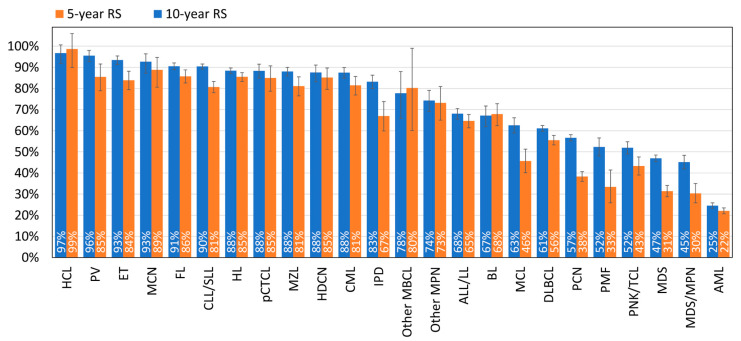
Five- and ten-year relative survival by HM type, Belgium. The blue and orange bars correspond to the 5-year and 10-year relative survival, respectively.

**Figure 5 cancers-15-04388-f005:**
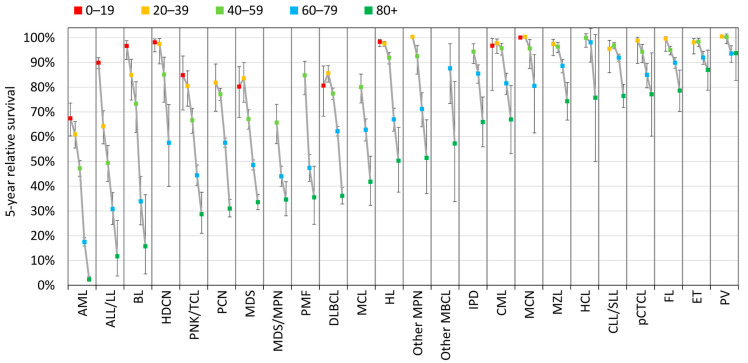
Age-specific 5-year relative survival by HM type and age group (years), Belgium 2009–2018.

**Figure 6 cancers-15-04388-f006:**
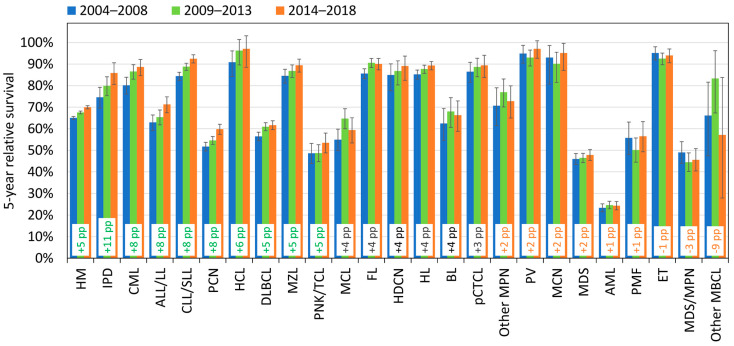
Five-year relative survival by cohort (2004–2008, 2009–2013, 2014–2018) and by HM type, Belgium (95% confidence intervals). The 24 HM subtypes are ordered according to the value of the percentage point (pp) differences in 5-year relative survival between cohorts 2004–2008 and 2014–2018; pp differences > 5 are indicated in green, those ≤ 2 in orange.

**Table 1 cancers-15-04388-t001:** Incidence, M/F ratio, and median age at diagnosis by HM type, Belgium 2018.

Haematological Malignancies (HMs)	N	CR	ESR 1976	ESR- 2013	WSR	WSR-Males	WSR-Females	M/F Ratio	Median Age
**All HMs**	7562	66.5	46.8	67.7	34.3	39.9	29.4	1.4	70
**Hodgkin Lymphoma (HL)**	364	3.2	3.1	3.2	3.0	3.4	2.6	1.3	40
**B-cell chronic lymphocytic leukaemia/small lymphocytic lymphoma (CLL/SLL)**	799	7.0	4.8	7.1	3.3	4.1	2.5	1.6	70
**Hairy cell leukaemia (HCL)**	55	0.5	0.4	0.5	0.3	0.5	0.1	8.1	64
**Other Mature B-cell leukaemia and related lymphoma (Other MBCL)**	18	0.2	0.1	0.2	0.0	0.1	0.0	3.0	79.5
**Immunoproliferative disease (IPD)**	192	1.7	1.1	1.7	0.7	1.0	0.5	2.1	73
**Plasma cell neoplasm (PCN)**	1009	8.9	5.8	9.1	3.8	4.7	3.1	1.5	72
**Marginal zone lymphoma (MZL)**	337	3.0	2.1	3.0	1.4	1.6	1.3	1.2	70
**Follicular lymphoma and related lymphoma (FL)**	413	3.6	2.7	3.7	1.9	2.0	1.8	1.1	65
**Mantle cell lymphoma (MCL)**	147	1.3	0.8	1.3	0.6	0.9	0.2	4.2	71
**Diffuse large B-cell lymphoma and related large B-cell lymphoma (DLBCL)**	875	7.7	5.2	7.8	3.7	4.2	3.3	1.3	71
**Burkitt lymphoma/leukaemia (BL)**	56	0.5	0.5	0.5	0.5	0.6	0.4	1.8	48
**Primary cutaneous T-cell lymphoma (pCTCL)**	108	0.9	0.7	1.0	0.6	0.8	0.3	2.5	66
**Peripheral NK/T-cell lymphoma (PNK/TCL)**	194	1.7	1.3	1.8	1.0	1.0	0.9	1.1	68
**Precursor lymphoid neoplasm or acute lymphoblastic leukaemia/lymphoma** **(ALL/LL)**	165	1.5	1.5	1.4	1.8	1.8	1.7	1.0	27
**Acute myeloid leukaemia and related precursor neoplasm** **(AML)**	611	5.4	3.8	5.4	2.8	3.0	2.7	1.1	69
**Chronic myeloid leukaemia (CML)**	158	1.4	1.1	1.4	0.9	0.9	0.8	1.2	64
**Polycythaemia vera (PV)**	222	2.0	1.4	2.0	1.0	1.2	0.8	1.5	69
**Essential thrombocythaemia (ET)**	418	3.7	2.6	3.7	1.9	1.6	2.1	0.7	68
**Primary myelofibrosis (PMF)**	104	0.9	0.6	0.9	0.4	0.5	0.3	2.0	73
**Other MPN and related neoplasm (Other MPN)**	41	0.4	0.2	0.4	0.2	0.2	0.1	2.0	73
**Mast cell neoplasm (MCN)**	38	0.3	0.3	0.3	0.3	0.2	0.3	0.6	49
**Myelodysplastic syndrome (MDS)**	848	7.5	4.4	7.6	2.8	3.5	2.3	1.5	77
**Myelodysplastic/myeloproliferative neoplasm (MDS/MPN)**	235	2.1	1.2	2.1	0.8	1.0	0.7	1.5	76
**Histiocytic and dendritic cell neoplasm (HDCN)**	40	0.4	0.4	0.4	0.4	0.6	0.2	3.6	40

N: Absolute number of new diagnoses; CR: crude incidence rate (N/100,000); ESR, ESR2013, WSR: age-standardised incidence rates using the European (1976 and 2013) or the world standard population (N per 100,000 person years); M/F ratio: male to female incidence ratio calculated using the WSR. Median age (in years) at diagnosis.

**Table 2 cancers-15-04388-t002:** Trends in age-standardised incidence (AAPC) by age group and HM type, males and females, Belgium 2004–2018.

Belgium 2004–2018	Males		Females		Males and Females	
**Trend by age group (years)**	**AAPC (%)**	**95%CI**	**AAPC (%)**	**95%CI**	**AAPC (%)**	**95%CI**
**0–14**	−0.1	[−1.7; 1.5]	0.4	[−1.3; 2.2]	0.1	[−1.2, 1.4]
**15–29**	1.0	[0.2; 1.9]	1.2	[−0.1; 2.5]	1.1	[0.3, 1.9]
**30–44**	1.1	[0.5; 1.7]	0.5	[−0.6; 1.6]	0.8	[0.3, 1.4]
**45–59**	0.6	[−0.3; 1.6]	1.1	[0.4; 1.7]	0.8	[0.1, 1.5]
**60–69**	1.3	[0.9; 1.8]	1.4	[1.0; 1.8]	1.4	[1.1, 1.7]
**70–79**	2.5	[2.1; 3.0]	3.0	[2.5; 3.6]	2.9	[2.4, 3.3]
**80+**	3.0	[2.2; 3.9]	3.1	[2.1; 4.0]	3.2	[2.4, 4.0]
**Trend by subtype**	**AAPC (%)**	**95%CI**	**AAPC (%)**	**95%CI**	**AAPC (%)**	**95%CI**
**HL**	1.2	[0.2; 2.3]	1.3	[0.4; 2.1]	1.2	[0.7; 1.8]
**CLL/SLL**	0.0	[−0.5; 0.4]	0.1	[−1.0; 1.2]	0.0	[−0.6; 0.6]
**HCL**	0.8	[−0.7; 2.4]	−1.6	[−6.0; 3.0]	−0.1	[−1.3; 1.0]
**IPD**	1.3	[0.0; 2.6]	1.9	[−0.1; 3.9]	1.2	[0.3; 2.1]
**PCN**	1.4	[0.8; 2.0]	1.0	[0.1; 2.0]	1.2	[0.6; 1.8]
**MZL**	3.1	[1.5; 4.7]	2.9	[1.5; 4.3]	2.1	[1.1; 3.2]
**FL**	0.0	[−0.6; 0.5]	0.2	[−0.9; 1.3]	0.1	[−0.6; 0.7]
**MCL**	0.3	[−0.8; 1.5]	−0.7	[−2.4; 1.0]	0.0	[−0.8; 0.9]
**DLBCL**	0.4	[−0.2; 1.0]	0.5	[0.0; 1.1]	0.5	[0.1; 0.9]
**BL**	1.6	[−1.3; 4.5]	1.2	[−4.8; 7.5]	1.4	[−0.9; 3.7]
**pCTCL**	0.8	[−0.5; 2.2]	−1.1	[−3.6; 1.4]	0.3	[−1.0; 1.6]
**PNK/TCL**	2.3	[0.7; 3.9]	3.9	[1.6; 6.2]	2.9	[1.5; 4.4]
**ALL/LL**	−0.1	[−2.1; 2.0]	−0.1	[−1.9; 1.7]	0.0	[−1.6; 1.5]
**AML**	1.1	[−0.1; 2.2]	1.4	[0.4; 2.6]	1.3	[0.4; 2.2]
**CML**	1.5	[0.0; 3.1]	1.5	[−0.9; 3.9]	1.5	[0.4; 2.6]
**PV**	6.3	[3.8; 8.9]	7.0	[5.9; 8.1]	6.0	[4.8; 7.2]
**ET**	4.8	[3.8; 5.7]	7.0	[5.6; 8.3]	6.0	[5.2; 6.7]
**PMF**	6.8	[4.9; 8.7]	4.8	[1.7; 8.0]	5.8	[4.5; 7.2]
**Other MPN**	1.2	[−4.2; 7.0]	−1.2	[−5.7; 3.6]	−0.8	[−5.4; 4.0]
**MCN**	9.7	[2.1; 17.9]	13.4	[9.2; 17.7]	11.2	[6.4; 16.3]
**MDS**	2.3	[1.2; 3.5]	3.8	[2.6; 5.0]	3.3	[2.4; 4.2]
**MDS/MPN**	3.4	[2.0; 4.8]	2.4	[0.5; 4.3]	3.0	[1.9; 4.2]
**HDCN**	2.7	[−0.7; 6.2]	−0.4	[−5.6; 5.0]	1.5	[−1.5; 4.7]

AAPC: average annual percentage change; 95%CI: 95% confidence intervals.

**Table 3 cancers-15-04388-t003:** Percentage point difference in 5-year relative survival between cohorts 2004–2008 and 2014–2018 by HM type and age group (years), Belgium.

	All Ages			0–14			15–19			20–39			40–59			60–79			80+		
HM Subtype	04–08	14–18	Δ pp	04–08	14–18	Δ pp	04–08	14–18	Δ pp	04–08	14–18	Δ pp	04–08	14–18	Δ pp	04–08	14–18	Δ pp	04–08	14–18	Δ pp
**IPD**	75%	86%	+11	-	-	-	-	-	-	-	-	-	88%	99%	+11	79%	87%	+9	39%	72%	+33
**CML**	80%	89%	+8	-	-	-	-	-	-	96%	99%	+3	93%	97%	+4	73%	84%	+11	46%	62%	+16
**ALL/LL**	63%	71%	+8	89%	93%	+4	69%	88%	+20	57%	74%	+17	44%	52%	+7	15%	37%	+22	-	-	-
**CLL/SLL**	84%	92%	+8	-	-	-	-	-	-	93%	95%	+1	93%	97%	+4	84%	94%	+10	69%	81%	+12
**PCN**	52%	60%	+8	-	-	-	-	-	-	-	-	-	71%	82%	+11	50%	61%	+11	27%	33%	+6
**HCL**	91%	97%	+6	-	-	-	-	-	-	-	-	-	102%	99%	−2	86%	105%	+18	-	-	-
**DLBCL**	56%	62%	+5	-	-	-	-	-	-	83%	88%	+5	71%	78%	+6	54%	64%	+9	30%	36%	+6
**MZL**	85%	89%	+5	-	-	-	-	-	-	96%	99%	+3	95%	98%	+3	83%	92%	+9	62%	70%	+8
**PNK/TCL**	49%	53%	+5	-	-	-	-	-	-	69%	83%	+14	55%	71%	+16	39%	48%	+9	27%	26%	−1
**MCL**	55%	59%	+4	-	-	-	-	-	-	-	-	-	74%	81%	+6	52%	59%	+7	28%	40%	+11
**FL**	86%	90%	+4	-	-	-	-	-	-	97%	99%	+2	93%	95%	+1	86%	90%	+4	49%	76%	+28
**HDCN**	85%	89%	+4	96%	100%	+4	-	-	-	-	-	-	-	-	-	-	-	-	-	-	-
**HL**	85%	89%	+4	99%	100%	+1	96%	96%	+0	96%	98%	+2	88%	93%	+5	60%	70%	+10	21%	66%	+45
**BL**	62%	66%	+4	94%	100%	+6	-	-	-	59%	80%	+21	-	-	-	31%	41%	+9	-	-	-
**pCTCL**	86%	89%	+3	-	-	-	-	-	-	96%	100%	+5	92%	94%	+3	82%	90%	+9	84%	59%	−25
**Other MPN**	71%	73%	+2	-	-	-	-	-	-	-	-	-	92%	91%	−1	67%	76%	+8	48%	42%	−6
**PV**	95%	97%	+2	-	-	-	-	-	-	-	-	-	100%	100%	+0	94%	95%	+1	86%	101%	+15
**MCN**	93%	95%	+2	-	-	-	-	-	-	-	-	-	-	-	-	-	-	-	-	-	-
**MDS**	46%	48%	+2	-	-	-	-	-	-	76%	88%	+13	56%	70%	+14	47%	50%	+3	34%	36%	+1
**AML**	23%	24%	+1	53%	70%	+17	-	-	-	53%	59%	+7	44%	47%	+3	14%	19%	+5	1%	2%	+1
**PMF**	56%	56%	+1	-	-	-	-	-	-	-	-	-	80%	86%	+7	51%	51%	+0	33%	46%	+13
**ET**	95%	94%	−1	-	-	-	-	-	-	99%	99%	+0	97%	100%	+3	95%	94%	−1	91%	83%	−8
**MDS/** **MPN**	49%	46%	−3	-	-	-	-	-	-	-	-	-	69%	64%	−5	48%	44%	−4	29%	38%	+9
**Other MBCL**	66%	57%	−9	-	-	-	-	-	-	-	-	-	-	-	-	-	-	-	-	-	-
**HM**	65%	70%	+5	89%	92%	+3	86%	90%	+4	85%	91%	+6	79%	85%	+6	61%	69%	+8	39%	47%	+8
Δ pp: Percentage point difference							Colour scale:			Δ pp 5–9			Δ pp 10–14			Δ pp 10–14			Δ pp 10–14		

## Data Availability

The data that support the findings of this study are available from info@kankerregister.org upon reasonable request in an aggregated format. The raw data are not publicly available due to privacy.

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
