# Peer review of "Epidemiological Trends of Haematological Malignancies in Belgium 2004–2018: Older Patients Show the Greatest Improvement in Survival"

_cancers, 2023, doi:10.3390/cancers15174388_

Round 1

Reviewer 1 Report

This manuscript provides unique Belgian population-based study concerning twenty four most common haematologic malignancies (98% of cases) over a fifteen-year period. A detailed analysis has been performed, in regard to incidence and its trends, survival and its trends, patients’ age and sex. In my opinion it is a model study which may give an impulse to extend it to European/worldwide. It moreover shows some directions where diagnostics/therapeutic procedures have to be improved as soon as possible. That seems particularly (but obviously not only) important for ageing societies.

Author Response

Thank you very much for this very positive review.

Reviewer 2 Report

Dear authors,

The paper is elegantly written and is easy to read, clear in its statement and qualifies for publication as long as the journal excepts this length of a manuscript without shortening.

However, I have two minor suggestions for improvement, namely 

(1) abbreviations in the text file as well as in tables should be consequently listed and clealy visible for the reader

(2) Tbl 1, 3 and S1 and Fig: 3 and 6 merit, to my opinion and for the sake of readabiliy, more explenatory comments/legends. 

Author Response

Thank you for your comments.

The detailed list of abbreviations of the different types of haematological malignancies has been added to Table 1.

The legends of figures and tables have been detailed, in particular the abbreviations have been better explained.

All tables have been reworked and additional comments have been added to figure and table legends as requested, including for Tables 1, 3 S1 and Figures 3 and 6.

Reviewer 3 Report

The objective of this study is to provide an insight into the epidemiological situation and trends of 24 main types of HM in Belgium during the last 15-year period, with a specific focus on the impact of age at the diagnosis that can be used to guide clinical decision making. They found that the incidence and survival rates increased, especially in the older population, and these increasing trends were explained by the diagnostic and therapeutic innovations over the last two decades. Authors concluded that the real-world study was useful to better monitor the impact of therapeutic changes on HMs at the population level.

In this analysis, the methodological approach is adequate, and the results obtained are precise and well explained. However, the illustrations are not clear enough and are probably taken from the Cancer Registry. Authors should create tables and figures uniformly and according to the guidelines of the journal.

The English should be improved by native speaker.

The English should be improved by native speaker.

Author Response

Thanks for your suggestions.

All illustrations have been reworked according to the guidelines of the Journal.

Additional comments have been added to figure and table legends.

The English language has been extensively revised by a native speaker.

Round 2

Reviewer 3 Report

I have no any other concerns.